# Single Crystal Growth and Physical Property Measurements of Sm$_3$Ti$X_5$ ($X$ = Bi, Sb)

Masahiro Shinozaki[1][*], Gaku Motoyama[1] Shijo Nishigori[2] Masahiro Tsubouchi[1] Kiyotaka Miyoshi[1] Kenji Fujiwara[1] and Masahiro Manago[1]

**1** Department of Material Science, Shimane University, Shimane 690-8504, Japan
**2** ICSR, Shimane University, Shimane 690-8504, Japan
* n20d102@matsu.shimane-u.ac.jp

August 16, 2022

**SCES Amsterdam 2022**

## Abstract

**We have performed synthesis of single-crystalline samples by flux method and physical properties measurements in Sm$_3$Ti$X_5$ ($X$ = Bi, Sb). Clear anomalies at $T_x$ around 15 K was observed in both systems. These anomalies at $T_x$ are very sharp and are accompanied by temperature hysteresis. As another feature, the phase transitions at $T_x$ hardly dependent on applying magnetic field at least up to 7 T. Although the detail of the low-temperature phase below $T_x$ is still unclear, it is suggested that these anomalies at $T_x$ are structural transition or valence transition of Sm ions from above behaviors.**

## 1 Introduction

Recently, some compounds with local inversion symmetry breaking have attracted great attention owing to the possibility as a field of odd-parity multipole ordering. The odd-parity multipoles can function as a origin of various unique properties such as cross-correlation phenomena. A zig-zag chain structure lacks a local inversion symmetry and is one of the ideal targets as a candidate of odd-parity multipole ordering [1, 2]. Actually, the magnetoelectric effect originated from magnetic toroidal dipole ordering has been reported on Ce$_3$TiBi$_5$ with Ce zig-zag chain structure [3, 4]. The series of compounds represented by $R_3TX_5$ ($R$: lanthanoid element, $T$: transition metal, $X$: $p$-block element) has many similar compounds and is expected to be a further research field for odd-parity multipole ordering. Synthesis of these compounds and the investigation of their physical properties are required. We have carried out single crystal growth and physical property measurements of Sm$_3$Ti$X_5$ ($X$ = Bi, Sb) in which

the physical properties are not clear in the present work [5, 6]. On the other hand, recently, it has been reported that some Sm compounds such as $SmOs_4Sb_{12}$ [7, 8] and $SmTi_2Al_{20}$ [9, 10] exhibits the magnetic field insensitive heavy fermion states. Hence, we have also focused on and investigated the behavior of physical properties of $Sm_3TiX_5$ in a magnetic field.

## 2 Experimental

Single-crystalline samples of $Sm_3TiBi_5$ were prepared by a Bi self-flux method. The purities of the materials of Sm, Ti, and Bi are 99.9%, 99.9%, and 99.99%, respectively. The starting materials were placed in the ratio Sm:Ti:Bi = 3:1:50 into a alumina crucible and sealed under high vacuum of $10^{-4}$ Torr in a quartz tube. The sealed ampoule was heated up 1000 °C, kept for 24 h, followed by a slow cool at 1.2 °C/h to 400 °C. The excess Bi flux was removed from crystals by using a centrifuge. Single-crystalline samples of $Sm_3TiSb_5$ were prepared by a Sn-flux method. The purities of the materials of Sb and Sn is 99.99% and 99.99%, respectively. The starting materials were weighed in the ratio Sm:Ti:Sb:Sn = 3:1:5:40, and crystals were grown in the same procedure as $Sm_3TiBi_5$. Obtained single crystals of $Sm_3TiSb_5$ were etched with 12 M HCl in 30 min in order to remove the residual Sn. The chemical composition determination of samples was carried out by the Energy Dispersive X-ray Spectroscopy (EDS) (Hitachi, Miniscope TM4000Plus). Powder X-ray Diffraction (XRD) was not performed for both systems in this work, because the total amount of obtained crystals was extremely small. Electrical resistivity and specific heat were measured by a standard four-terminal method and a conventional adiabatic heat pulse method. Magnetization measurements were performed using a commercial SQUID magnetometer (Quantum Design, MPMS).

## 3 Results and discussion

Single crystal specimens of $Sm_3TiX_5$ ($X$ = Bi, Sb) were successfully grown. The chemical composition ratio of obtained crystals was confirmed as approximately Sm:Ti:$X$ = 3:1:5 by EDS measurements. Maximum size of obtained crystals is 1.5×(1.0×1.0) $mm^3$ for $Sm_3TiBi_5$. However, all obtained single crystal specimens of $Sm_3TiSb_5$ were very tiny. Typical size and mass of samples of $Sm_3TiSb_5$ are 0.8×(0.1×0.1) $mm^3$ and 0.0002 g, respectively. In order to maintain enough measurement accuracy, only electrical resistivity measurement was performed for $Sm_3TiSb_5$ in our present work. Additionally, we have also tried single crystal growth of $Sm_3TiSb_5$ by using other elements as flux: Al, Zn, Ga, In, Pb, and Sb (self-flux). As a result, no single crystals were obtained except for Sn-flux under the preparation condition described in the experimental chapter.

## 3.1  $\mathbf{Sm_3TiBi_5}$

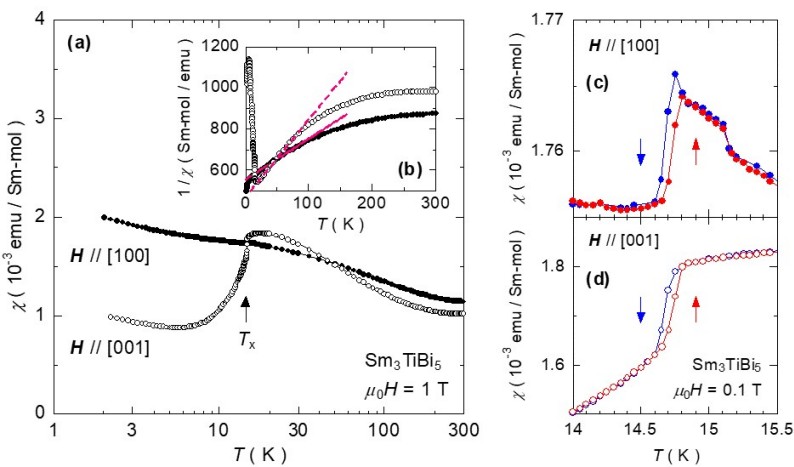

Figure 1: Temperature dependences of **(a)** magnetic susceptibility and **(b)** reciprocal susceptibility of $Sm_3TiBi_5$ at $\mu_0 H = 1$ T. The solid and dashed lines indicate Curie-Weiss fitting within $T = 30$ K to 80 K in Fig. 1**(b)**. Temperature dependences of magnetic susceptibility of $Sm_3TiBi_5$ near $T_x$ in the magnetic field direction along **(c)** [100] and **(d)** [001].

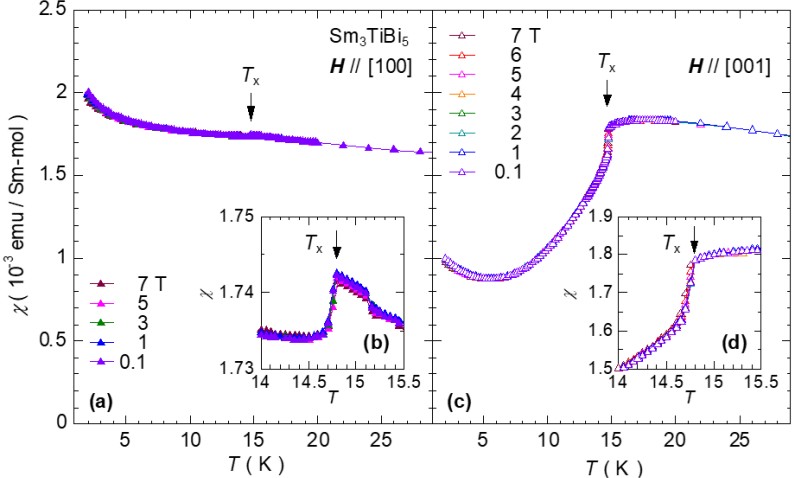

Figure 2: **(a) (b)** Temperature dependences of magnetic susceptibility of $Sm_3TiBi_5$ below 30 K at applying several magnetic field. **(c) (d)** Inset shows an enlarged view near $T_x$.

Figure 1(a) shows temperature $T$ dependence of magnetic susceptibility $\chi$ of $Sm_3TiBi_5$ at 1 T. The anisotropy of $\chi(T)$ between $H$ // [100] and [001] is small in the paramagnetic state from $T = 300$ K to around 20 K, and $\chi(T)$ gradually increases with decreasing temperature. The characteristic of $\chi(T)$ on the high $T$ region is that $\chi(T)$ does not follow Curie-Weiss law as shown Fig. 1(b). Temperature dependences of reciprocal magnetic susceptibility in both directions do not follow $T$-linear dependence over a wide $T$ range. This suggests that the valence of Sm ion changes with temperature change at least in this $T$ range. The effective magnetic moments $\mu_{eff}$ and Weiss temperature $\Theta_p$ are estimated by the fitting to the Curie-Weiss law within the lower $T$ ranges 30 K to 80 K. The formula $\chi = (T-\Theta_p)/C$ is used for the fitting function, where $C$ is the Curie constant. Obtained $\mu_{eff}$ values are 2.01 $\mu_B$ and 1.46 $\mu_B$

70  for the [100] and [001] directions, respectively. The $\mu_{eff}$ value in the [001] direction is close to
71  the expected value of free $Sm^{3+}$ ion including the Van Vleck contribution (1.53 $\mu_B$). However,
72  the $\mu_{eff}$ value in the [100] direction takes a value between $Sm^{3+}$ and $Sm^{2+}$ (3.40 $\mu_B$). $\Theta_p$ are
73  obtained as –280 K and –125 K for the [100] and [001] directions, respectively. This suggests
74  an effective antiferromagnetic (AFM) interaction between the $4f$ electrons. In low $T$ region of
75  Fig. 1(a), a clear anomaly at $T_x = 14.8$ K can be observed. Although gradually increasing of
76  $\chi(T)$ toward $T = 0$ K is common feature, there is a large anisotropy of $\chi(T)$ between $H$ // [100]
77  and [001] below $T_x$. $\chi(T)$ in $H$ // [100] is almost constant and gradually increases below $T_x$.
78  In contrast, $\chi(T)$ in $H$ // [001] is after being suddenly suppressed just below $T_x$, then it also
79  gradually increases. These features below $T_x$ resemble $\chi(T)$ in an AFM ordered state in which
80  the ordered magnetic moments orient in (001) plane. Figure 1(c) and (d) show $\chi(T)$ around
81  $T_x$. Clear kinks of $\chi(T)$ can be confirmed, and temperature hysteresis behavior is also observed
82  at $T_x$ in both directions. Thus, the anomaly at $T_x$ is considered something first-order phase
83  transition. Figure 2 show $\chi(T)$ under various applied magnetic field condition at low $T$ region.
84  Both the absolute value of $\chi$ and $T_x$ hardly change for the magnitude of magnetic field and
85  are seen to maintain almost same value at zero filed. These features indicate a possibility of
86  some ordered state different from the ordinally AFM ordering which is able to be suppressed
87  by the magnetic field. From the constant $\chi$ in spite of increasing of magnetic field, the low-$T$
88  state below $T_x$ is considered to be paramagnetic state. It is considered that the anomaly at $T_x$
89  is a valence transition of Sm ions or a structural transition, because the phase transition at $T_x$
90  is non-magnetic first-order transition.

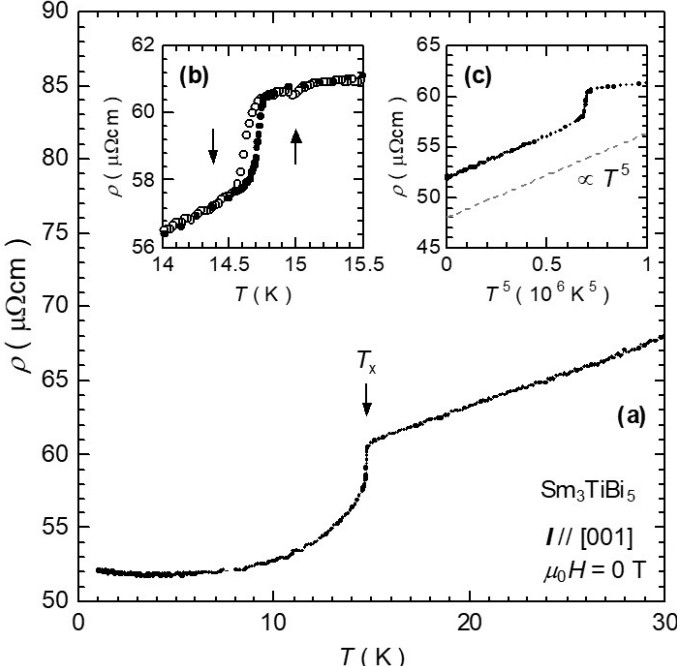

Figure 3: **(a)** Temperature dependence of electrical resistivity of $Sm_3TiBi_5$ at zero magnetic field. **(b)** An enlarged view near $T_x$. The solid and open circles are corresponding to heating and cooling, respectively. **(c)** $\rho$ vs $T^5$ plots below 16 K.

91   Next, temperature dependence of electrical resistivity $\rho$ of $Sm_3TiBi_5$ under zero magnetic
92  field condition is shown in Fig. 3. Clear kink of $\rho(T)$ is observed at phase transition tem-
93  perature $T_x = 14.8$ K which is same temperature estimated by magnetization measurements.
94  $\rho(T)$ rapidly decreases with decreasing $T$, then becomes almost constant value. This rapidly
95  decreasing of $\rho(T)$ below $T_x$ can be considered due to the decreasing of conduction electron

96  with the valence change of Sm ion from $Sm^{3+}$ or intermediate valence to $Sm^{2+}$. Temperature
97  hysteresis observed in $\chi(T)$ at $T_x$ also clearly exhibits even in $\rho(T)$ as shown Fig. 3(b). In Fig.
98  3(c), it is shown that $\rho(T)$ below $T_x$ exhibits $T^5$ dependence and follows the formula: $\rho(T)$
99  $= \rho_0 + AT^2 + \beta T^5$, where $\rho_0$ is residual resistivity, and $A$ is a coefficient of electron-electron
100 interaction. $T^5$ term is phonon contribution term in the sufficiently lower $T$ range than the
101 Debye temperature $\Theta_D$ ($T \ll \Theta_D/2$). $\Theta_D$ of $Sm_3TiBi_5$ is estimated $\Theta_D = 64$ K below $T_x$ from
102 the following specific heat data in $T = 4$ to 12 K. $T^5$ term is dominant part of $\rho(T)$ in the low-$T$
103 phase. This result suggests that the electron correlation becomes sufficiently weak below $T_x$.
104 Additionally, this is consistent with the scenario in which conduction electrons decrease with
105 valence transition of Sm ion, although there is still the possibility that the phase transition at
106 $T_x$ is the structural transition.

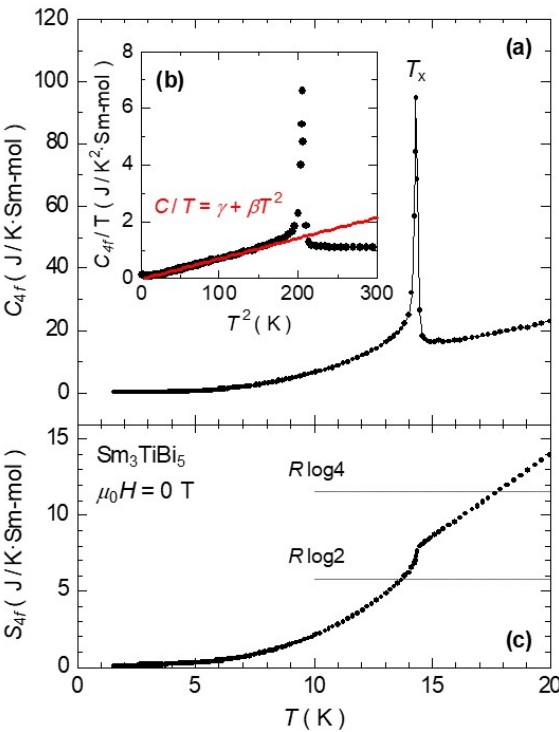

Figure 4: **(a)** Temperature dependence of specific heat of $Sm_3TiBi_5$ at zero magnetic
field. Only the 4$f$ electron contribution component is plotted in this graph. **(b)**
$C_{4f}/T$ vs $T^2$ plot of the result of Fig. 4(a). The solid line indicates a $T^2$ fitting. **(c)**
Temperature dependence of magnetic entropy estimated from the result of Fig. 4(a).

107 Figure 4(a) shows temperature dependence of specific heat $C_{4f}$ of $Sm_3TiBi_5$ under zero
108 magnetic field condition. $C_{4f}$ is subtracted a lattice contribution by subtracting the specific
109 heat of $La_3TiBi_5$ from the raw data. A symmetric and very sharp peak was observed at $T_x$,
110 which is different from a jump of $C(T)$ at second-order phase transition. $T_x$ in Fig. 4(a) is
111 14.3 K and slightly small value in comparison with that in $\chi(T)$ and $\rho(T)$. From $C_{4f}/T$ vs $T^2$
112 curve as shown in Fig. 4(b), the Sommerfeld coefficient $\gamma$ below $T_x$ is estimated to be almost
113 zero within the measurement accuracy. Curve fitting was performed using the fitting formula:
114 $C/T = \gamma + \beta T^2$, where $\beta$ is a phonon contribution coefficient. Because $\gamma$ is very small value,
115 $Sm_3TiBi_5$ does not considered to be a heavy fermion system and the electron correlation is
116 sufficiently weak below $T_x$. This is consistent with the above result about the temperature
117 dependence of $\rho$ below $T_x$. Thus, although the ground state of $Sm_3TiBi_5$ is hardly dependent
118 phase for magnetic field, it is not field-insensitive heavy fermion state as reported in $SmOs_4Sb_{12}$
119 and $SmTi_2Al_{20}$. Fig. 4(c) shows the entropy $S_{4f}$ of $Sm_3TiBi_5$ estimated from the result of $C_{4f}$.

120  $S_{4f}$ at $T_x$ reaches about 70% of $R\ln 4$, which indicates that the CEF ground state of $Sm_3TiBi_5$ is
121  quartet.

## 3.2  $Sm_3TiSb_5$

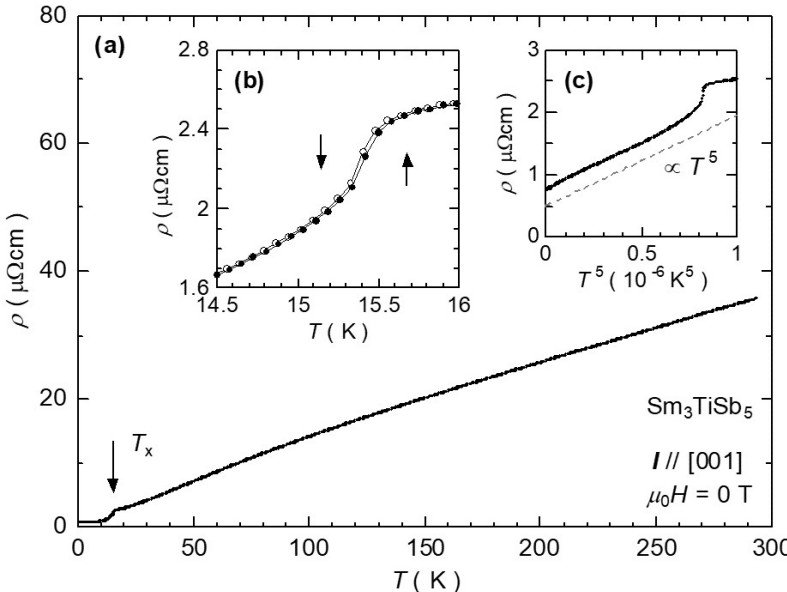

Figure 5: **(a)** Temperature dependence of electrical resistivity of $Sm_3TiSb_5$ at zero
magnetic field. **(b)** An enlarged view near $T_x$. The solid and open circles are corre-
sponding to heating and cooling, respectively. **(c)** $\rho$ vs $T^5$ plots below 16 K.

123  Figure 5(a) shows temperature dependence of electrical resistivity of $Sm_3TiSb_5$ at zero mag-
124  netic field condition. In the high temperature region above 20 K, there are no anomaly and
125  $\rho(T)$ curve shows a $T$-linear like dependence with slightly upwardly convex. This slight curva-
126  ture of $\rho(T)$ is considered to be due to the change of the valence of Sm ion with temperature
127  change, similar to that suggested in magnetic susceptibility of $Sm_3TiBi_5$. $\rho(T)$ exhibits a clear
128  kink at $T_x = 15.5$ K, then $\rho(T)$ only decreases with decreasing temperature. $\rho(T)$ saturates
129  near 5 K and has a constant value at lower temperature. The superconducting transition of
130  residual Sn, which is often seen in $RE_3TiSb_5$ ($RE$: lanthanoid element) systems grown by Sn-
131  flux method [6, 11], is not confirmed. Hence, our samples seem to be a high purity single
132  crystal with no residual Sn. The residual resistivity $\rho_0$ at 5 K is 0.76 $\mu\Omega$ cm, and the residual
133  resistivity ratio $RRR$ ($= \rho_{300K}/\rho_{5K}$) is about 48. The absolute value of $\rho(T)$ is very smaller
134  than that of previous research, and the $RRR$ is about 6 times larger. These features also guar-
135  antee our sample quality. A small temperature hysteresis emerges on $\rho(T)$ at $T_x$ as shown
136  in Fig. 5(b). Although the temperature hysteresis of $\rho(T)$ of $Sm_3TiSb_5$ is smaller than that
137  of $Sm_3TiBi_5$, as in the case of $Sm_3TiBi_5$, these clear kink and temperature hysteresis are sug-
138  gested that the anomaly at $T_x$ is something first-order phase transition. $\rho(T)$ below $T_x$ seems
139  to follow the $T^5$ dependence as shown Fig. 5(c) similar to the case of $Sm_3TiBi_5$. Finally, the
140  magnetic field dependence of $\rho(T)$ of $Sm_3TiSb_5$ is shown in Fig. 6. It can be seen that $\rho(T)$
141  almost overlaps one another between zero magnetic field and 9 T in both directions of $H$ //
142  [100] and [001]. Although there is a little magnetic resistance, $T_x$ are not change from $T_x =$
143  15.5 K over the all magnetic field range. These indicate that the phase transition at $T_x$ and the
144  low-temperature phase below $T_x$ in $Sm_3TiSb_5$ are robust to the external magnetic field just
145  like those of $Sm_3TiBi_5$.

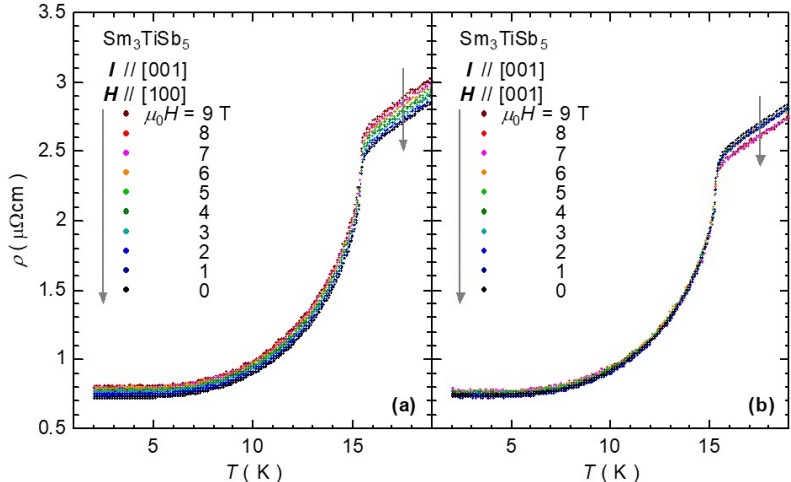

Figure 6: Temperature dependences of electrical resistivity of $Sm_3TiSb_5$ at applying several magnetic field up to 9 T along the direction **(a)** [100] and **(b)** [001].

## 4  Conclusion

We have carried out single crystal growth and physical property measurements of $Sm_3TiX_5$ ($X$ = Bi, Sb). Clear kink at $T_x$ around 15 K was successfully observed in both compounds. Since these kinks at $T_x$ are very sharp and show the temperature hysteresis, it is suggested that these are something first-order phase transition. However, the detail of the low temperature phase below $T_x$ is still unclear. We have also revealed that $T_x$ is hardly dependent on applied magnetic field at least up to 7 T. Following above features, there is a possibility that the anomalies at $T_x$ in $Sm_3TiX_5$ is structural transition or valence transition of Sm ions.

## Acknowledgements

This work was supported by the technical staff at ICSR, Shimane University. The authors thank T. Matsumoto for experimental help. One of the starting materials of 4N purity Sb was provided by NIHON SEIKO Co., Ltd.

**Funding information**   This work was supported by JSPS KAKENHI Grant No. 21K03447.

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
