# Peer review of "Single Crystal Growth and Physical Property Measurements in Sm3TiX5 (X = Bi, Sb)"

_SciPost Physics Proceedings_

## Round 1 · Referee Report · Anonymous (Referee 1) · 2022-10-31

Report

In this paper, the authors have successfully grown single crystal samples of Sm_3TiX_5 (X = Bi, Sb) using the self-flux method with each X element and report the results of basic physical property measurements.
Although the paper is well written and some experimental results have progressed from previous reports, however, some of the following points are lacking. If the authors are resubmitting the paper, please reconsider the following points.

Requested changes

1. The lack of XRD results gives a negative impression. The authors checked the composition ratio with EDS, how accurate is it? If the spatial mapping of elemental composition ratios in the single crystal surface has eliminated the possibility of a binary phase of Bi or Sb, please mention it.

2. Anyway, please indicate the crystal structure and space group of these compound.

3. How did authors identify the 100 and 001 directions in Fig. 1? If the Laue method were used, the author should be able to mention the crystal structure and lattice constants from the analysis of Laue spots. (If it is analogous to past research results of similar materials such as the Ce system and its natural facets, it should be written as such.)

4. The authors mention CEF ground state in the system X=Bi, if you are assuming Sm^{3+} (J = 5/2) state, please specify. If the authors are going to assert like ”…is quartet”, more information about the site symmetry of Sm and the how to split the J multiplet.

5. By the way, in the case of hexagonal system, and considering the double groups, they will all split into Kramers doublet, so there can't be a quartet. Do the authors intend accidental degeneracy and/or pseudo-degeneracy? Does the quadrupolar degrees of freedom contribute to the structural transition?

6. Are the arrows on the right panel of Fig. 6 in the Graph legend and the arrows on the data reversed? For me, the purple data appears to have smaller resistivity than the blue data at around 17.5 K.

Minor points
1. It is easier to compare with other compounds if \mu_{eff} is described as a value per one Sm ion.
2. If the valence of Sm varies with temperature, it is better to refer to a typical material that has been confirmed by synchrotron experiments in the past. For example, SmOs4Sb12.
3. The arrows pointing in the direction of temperature sweep in the inset of Fig. 1 (d), Fig. 3(b), and Fig. 5(b) should be a little closer to the data and slightly tilted to match the slope of the data. If it is displayed as perpendicular to the temperature axis, it will be confused with the arrow pointing to some anomaly.

---

## Editorial Decision

awaiting_resubmission